# MSDS: A Large-Scale Chinese Signature and Token Digit String Dataset for Handwriting Verification

**Peirong Zhang**
South China University of Technology
eeprzhang@mail.scut.edu.cn

**Jiajia Jiang**
South China University of Technology
eejiajia_jiang@mail.scut.edu.cn

**Yuliang Liu**
Huazhong University of Science and Technology
ylliu@hust.edu.cn

**Lianwen Jin**[*]
South China University of Technology
eelwjin@scut.edu.cn

## Abstract

Although online handwriting verification has made great progress recently, the verification performances are still far behind the real usage owing to the small scale of the datasets as well as the limited biometric mediums. Therefore, this paper proposes a new handwriting verification benchmark dataset named Multimodal Signature and Digit String (MSDS), which consists of two subsets: MSDS-ChS (Chinese Signatures) and MSDS-TDS (Token Digit Strings), contributed by 402 users, with 20 genuine samples and 20 skilled forgeries per user per subset. MSDS-ChS consists of handwritten Chinese signatures, which, to the best of our knowledge, is the largest publicly available Chinese signature dataset for handwriting verification, at least eight times larger than existing online datasets. Meanwhile, MSDS-TDS consists of handwritten Token Digit Strings, i.e, the actual phone numbers of users, which have not been explored yet. Extensive experiments with different baselines are respectively conducted for MSDS-ChS and MSDS-TDS. Surprisingly, verification performances of state-of-the-art methods on MSDS-TDS are generally better than those on MSDS-ChS, which indicates that the handwritten Token Digit String could be a more effective biometric than handwritten Chinese signature. This is a promising discovery that could inspire us to explore new biometric traits. The MSDS dataset is available at https://github.com/HCIILAB/MSDS.

## 1  Introduction

Handwritten signature is a biometric measure that has been profoundly exploited in identity verification and related studies have rapidly progressed in recent years [4]. Signature verification is to authenticate a tested signature by comparing it to the template of its stated authorship. Owing to the large intra-writer variance in human handwriting (Figure 1), a signature verification system is easily attacked by skilled forgeries from malicious forgers. Therefore, handwritten signature verification is still challenging. According to the manner of data acquisition, signatures can be categorized into online and offline modalities [4]. Online signatures are recorded by electronic devices with temporal, positional, and pressure information and are stored as time series, whereas offline signatures are acquired from static signature images by photographing or scanning.

So far, considerable research has been conducted on signature verification, such as Dynamic Time Warping (DTW) [43, 19, 31, 35, 2, 28], Hidden Markov Models (HMM) [9, 8, 26, 6] for online verification and hand-crafted features [32, 5, 32, 14, 17, 1, 20], deep features [10, 23, 22, 48] for

---

[*]Corresponding author.

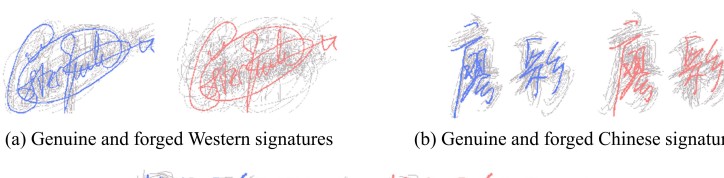

(a) Genuine and forged Western signatures    (b) Genuine and forged Chinese signatures

(c) Genuine and forged Token Digit Strings

Figure 1: The intra-class variance of handwriting, including English signatures from DeepSignDB [40], Chinese signatures from MSDS-ChS, and Token Digit Strings from MSDS-TDS. The ones marked in red and blue are the same genuine samples of the specific user, and the ones marked in gray are the other samples. The left column denotes the intra-class variance of all genuine samples, whereas the right column denotes the intra-class variance of all skilled forgeries.

offline verification. However, the datasets used in most signature verification researches were not in Chinese, but in English, Hungarian, Dutch, and other languages [15, 47, 30, 11, 24, 37, 40]. DeepSignDB [40], the largest online signature database in the Western language, contains 1526 users and almost 70,000 samples. But the existing public online Chinese signature datasets [25, 24, 47] are considerably smaller, with no more than 50 registered users, hindering researchers from in-depth explorations of Chinese signature verification.

Handwritten digits and characters are also important behavioral biometric traits, similar to handwritten signatures. However, existing digit/character-related datasets [38, 39, 3] have several limitations. *(1)* Most of them collected data in a single digit or letter way, thus failing to capture continuous writing information, which contains richer personal writing styles. *(2)* Although some considered the combination of single-digit, such a combination without natural connections between digits is essentially different from naturally handwritten digit strings. In addition, writers are unfamiliar with the digit combinations, resulting in weaker muscle memory and the loss of handwriting style. Consecutive digit strings that we usually handwrite in reality include phone numbers, ID numbers, etc. They are strongly unique and exclusive and we define them as Token Digit Strings (TDS). As we are fairly familiar with them and form strong muscle memories about writing them, the TDS contains rich discriminative personal writing characteristics and is a potential identity-verification medium. A new TDS dataset is needed for further study, as there are no such datasets yet.

To facilitate related research and inspire future work, we establish the Multimodal Signature and Digit String (MSDS) dataset, a multimodal online and offline handwriting dataset that comprises two subsets: MSDS-ChS and MSDS-TDS, where the former contains handwritten Chinense signatures and the latter contains handwritten TDS. A total of 402 users have contributed their handwriting, with 20 genuine samples and 20 skilled forgeries for each subset. The data were acquired in two captured sessions with a time gap of at least 21 days. Online handwriting information and the corresponding offline images are jointly provided. In addition, we carried out a comprehensive benchmark evaluation and jointly analyzed the two subsets.

To summarize, the major characteristics of MSDS dataset are as follows:

- MSDS-ChS is the largest publicly available Chinese signature dataset, whose size is at least eight times larger than the previous online Chinese signature datasets [25].
- MSDS-TDS is the first large-scale Token Digit String dataset that studies the effectiveness of handwritten TDS. It aims to explore a new and more efficient biometric for identity verification, which brings long-term implications for related research fields.
- MSDS has taken into account inter-session variation of the handwriting from the same user by acquiring data in two separate sessions. This simulates the real-world scenarios to conduct a more valid assessment, which enhances the feasibility of our dataset.

To evaluate the effectiveness of MSDS, we design a thorough benchmark evaluation with different baselines, including the state-of-the-art DsDTW [13], TA-RNNs [40], etc. Additionally, we conduct a modality fusion evaluation to investigate the improvement of fusing online time series and offline images and a cross-dataset validation to justify the requirement for a large-scale Chinese signature dataset. Experimental results on MSDS-TDS are generally better than those on MSDS-ChS, which reveals with surprise that the Token Digit String is more powerful than Chinese signature. This

interesting and important finding inspires us that we can adopt TDS instead of Chinese signatures for high-accurate online handwriting verification in real-world applications.

## 2    Related Works

In the past decades, numerous online and offline handwriting verification datasets have been published in the literature, most of which are related to signatures whereas few of them cover digits/digit strings.

### 2.1    Signature Datasets

*(A) Online Signature Datasets*

In 2003, Ortega et al. [29] published the MCYT database, in which a subcorpus was signature-based. This subcorpus contains signatures from 330 writers, each of whom wrote 25 genuine signatures and 25 forgeries. Yeung et al. [47] proposed the SVC dataset, a mixed language dataset containing both Chinese and English. Its signatures are not actual names and there are only 40 users included. Regarding various input scenarios, Tolosana et al. [37] recorded the signatures written by finger and stylus on different COTS (commercial off-the-shelf) devices in the e-BioSign database. They evaluated the verification performance under three scenarios: intra-device, inter-device, and mixed writing-tool. Lu et al. [25] proposed SCUT-MMSIG, a Chinese signature dataset that possesses three modalities: mobile, tablet, and in-air. 50 writers wrote 20 genuine and forged signatures in each modality, respectively. Tolosana et al. [40] proposed DeepSignDB, which was contributed by a total of 1,526 users with different numbers of signatures written in finger and stylus scenarios, and is the largest online handwriting western signature dataset up-to-date. SVC2021_EvalDB [41] was a dataset specifically acquired for final evaluation of SVC2021. It contains signatures from 75 subjects collected in the office scenario in two sessions and 119 subjects collected in the mobile scenario in four to six sessions.

*(B) Offline Signature Datasets*

The MCYT dataset mentioned above had a subset of 75 subjects, a.k.a MCYT-75, specifically for the usage in offline scenarios. Kalera et al. [15] proposed the CEDAR dataset that consists of 55 users with 24 genuine samples per writer. They also collected 24 forged samples for each user written by other 20 skillful writers. SigWIComp2015 [27] used in the ICDAR2015 Competition includes the Italian Offline Signatures and Bengali Offline Signatures subsets. The signatures in the former subset were actual signatures obtained from forms and applications of students and were collected over three to five years. Soleimani et al. [33] published the UTSig dataset in the language of Persian. Among 45 forged samples of each class, three are written by opposite-hand, which improves the model's performance. Pal et al. [30] proposed the BHSig260 dataset, an Indic-script signature dataset in Bengali and Hindi. Among a total of 260 users, 100 users wrote in Bengali and the other 160 ones wrote in Hindi, with 24 genuine signatures and 30 skilled forgeries per user. Yan et al. [46] proposed ChiSig, a signature forgery detection benchmark that contains 10,242 samples from 102 users. In addition, there are two non-public Chinese signature datasets. Hu et al. [12] proposed an offline Chinese signature dataset that possesses 300 users and 5400 samples. Wei et al. [44] published the CSD dataset, which includes a total of 749 names and approximately 29,000 signature images.

### 2.2    Single Character/Digit/Letter Datasets

Although handwritten digits and digit strings imply masses of distinctive personal writing features, they have not attracted the attention of researchers in verification as significant as signatures. To enhance the One-Time Passwords authentication systems, Tolosana et al. [38] published the e-BioDigit database, which consists of online handwritten digits from 0 to 9 written by finger on mobile devices and would serve as second-level identity authentication. Then, in their following work [39], they presented the new MobileTouchDB public database. Users not only wrote separate digits but also the same 4-digit passwords by finger. Chen et al. [3] proposed the LERID database, which is composed of English single-letter.

Compared with the aforementioned online Chinese signature dataset, our proposed MSDS-ChS subset is more than eight times larger than the one with the largest number of users (SCUT-MMSIG[25], 50 users). The MSDS-TDS subset firstly covers handwritten Token Digit String, which is more common than separate digits and worthy of in-depth exploration. Furthermore, MSDS considers the inter-session variation of the handwriting from the same writer, which could be ignored in previous datasets, assessing the value of these two types of handwriting in more realistic scenarios.

# 3 MSDS Dataset

MSDS is a novel dataset that contains two subsets: MSDS-ChS for handwritten Chinese signatures and MSDS-TDS for handwritten Token Digit Strings (TDS). The two subsets are contributed by the same 402 users with the same protocol. For data acquisition, we used two devices: HUAWEI MatePad BAH3-W59 and LENOVO TB-J706F, with three of each. Both are Android-based tablets and have specific stylus of their own. The specifications of the two devices are included in the supplementary materials. We specifically developed an Android app, and the user interface is depicted in Figure 2, which is composed of the main writing board, progress bar, toolbar, and information display area. The users wrote their signatures, as shown in Figure 2(a), and TDS, as shown in Figure 2(b), on the writing board.

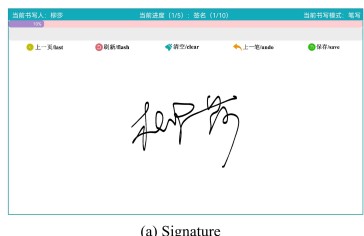
(a) Signature

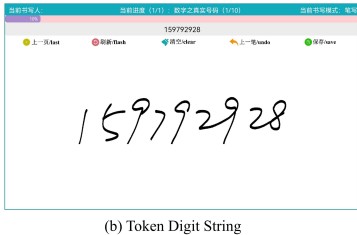
(b) Token Digit String

Figure 2: The user interface of the data acquisition app.

The data acquisition process is divided into two separate sessions with a time interval of at least 21 days. In each session, users performed writing according to the same procedure: 10 genuine signatures→10 genuine phone numbers→10 forged signatures→10 forged phone numbers. When performing genuine phone numbers, the user was allowed to write a previously used phone number but must ensure that this phone number was extremely familiar. When performing skilled forgeries, the imitator imitated the handwriting of another random-picked user by repeatedly watching the screen recording of the writing process and practicing. To avoid duplication, the imitator and imitated user are uniquely corresponding. Hence, after two sessions finished, each user contributed a total of 20 genuine/forged signatures and 20 genuine/forged phone numbers. The dynamic information recorded during the writing process includes $x, y$ coordinates, pressure, and time stamps, which are saved in separate text files. In addition, we saved the corresponding static images of each handwriting in the Portable Network Graphics (PNG) format. Samples of handwriting are shown in Figure 3 and more samples are included in supplementary materials. The sizes of static images vary because of the distinct screen sizes of two kinds of acquisition devices.

The users come from various cities/provinces of China, which presents a rich regional diversity. Regarding age, they are between 20 and 28. Previous literature [36, 7] has shown that the main characteristics of an adult's handwriting are not influenced by age. Therefore, the users' handwriting is largely mature and stable as they are all adults. Regarding gender, 60.2% of the contributors are males and 39.8% are females. Before collecting users' handwriting data, we signed a copyright agreement with each of them, in which they gave their consent to the data collection and agreed to grant us the license to use their handwriting for non-commercial academic research purposes and publication. Besides, users have the right to withdraw their handwritten data from the dataset.

We summarize the MSDS dataset in Table 1 and present comparisons with existing datasets from different aspects in Table 2 and 3, including data modality, the number of users and samples, etc. When providing the data, we respectively shuffle the user order of Chinese signatures and TDS, resulting in an unassociated order between Chinese signatures and TDS.

Table 1: A summary of the proposed MSDS dataset.

| Subset | Content | Modality | | User | Genuine Sample[1] | Skilled Forgery[1] | Features[2] |
|---|---|---|---|---|---|---|---|
| | | online | offline | | | | |
| MSDS-ChS | Chinese Signature | ✓ | ✓ | 402 | $402 \times (10+10) = 8,040$ | $402 \times (10+10) = 8,040$ | $X,Y,P,T,I_r$ |
| MSDS-TDS | Token Digit String | ✓ | ✓ | 402 | $402 \times (10+10) = 8,040$ | $402 \times (10+10) = 8,040$ | $X,Y,P,T,I_r$ |

[1] Each user contributed 10 samples in two sessions.
[2] $X,Y,P,T,I_r$ respectively denote the $x, y$ coordinates, pressure, timestamps, and static rendered images.

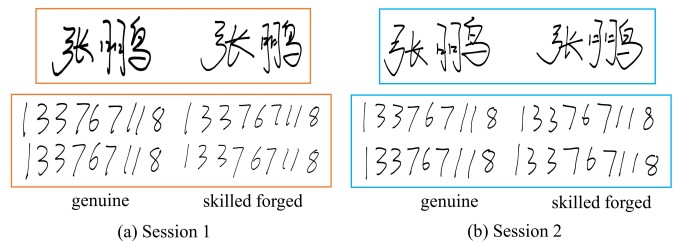

|   | genuine | skilled forged |   | genuine | skilled forged |
| (a) Session 1 | | | (b) Session 2 | | |

Figure 3: Signature and TDS samples of two sessions. The left ones are genuine samples, whereas the right ones are skilled forgeries.

Table 2: Comparisons with existing Chinese signature datasets.

| Dataset (subcorpus) | Modality | | User | Genuine/Forged Sample | Sessions | Session Interval | Features[3] |
| | online | offline | | | | | |
|---|---|---|---|---|---|---|---|
| SVC2004-Task1 [47] | ✓ | ✗ | 40 | 800/800 | 1 | - | $X, Y$ |
| SVC2004-Task2 [47] | ✓ | ✗ | 40 | 800/800 | 1 | - | $X, Y, P, A_z, A_t, T, B$ |
| SigComp 2011 [24] | ✓ | ✓ | 20 | 471/707 | 1 | - | $X, Y, Z, I_s$ |
| SCUT-MMSIG [25] | ✓ | ✗ | 50 | 3,000/3,000[1] | 2 | 1-3 months | $X, Y, T, B$ |
| Dataset in [12][2] | ✗ | ✓ | 300 | 5,400 | 1 | - | $I_s$ |
| CSD [44][2] | ✗ | ✓ | 749 | 29,000 | 1 | - | $I_s$ |
| ChiSig [46] | ✗ | ✓ | 102 | 10,242 | 1 | - | $I_s$ |
| MSDS-ChS (Ours) | ✓ | ✓ | 402 | 8,040/8,040 | 2 | ≥21 days | $X, Y, P, T, I_r$ |

[1] There are 1000 samples in three modes each: mobile, tablet, and in-air, respectively.
[2] Non-public.
[3] $X, Y, Z, P, T, I_r, I_s, A_z, A_t, B$ respectively denote the $x, y, z$ coordinates, pressure, timestamps, static rendered images, scanned images, azimuth, altitude, and button status.

Table 3: Comparisons with existing character/digit/letter-based datasets.

| Dataset (subcorpus) | Content | User | Genuine/Forged Sample | Sessions | Session Interval | Features[1] |
|---|---|---|---|---|---|---|
| MobileTouchDB [39] | Character | 217 | 64K/- | 6 | ≥2 days | $X, Y, T, S_f, A_{cc}, G$ |
| e-BioDigitDB [38] | Digit | 93 | 7,440/- | 2 | ≥21 days | $X, Y, P, T$ |
| LERID [3] | Letter | 414 | 107,723/- | - | - | $X, Y$ |
| MSDS-TDS (Ours) | TDS | 402 | 8,040/8,040 | 2 | ≥21 days | $X, Y, P, T, I_r$ |

[1] $X, Y, P, T, S_f, I_r, A_{cc}, G$ respectively denote the $x, y$ coordinates, pressure, timestamps, the area covered by the finger, static rendered images, accelerometer, and gyroscope.

# 4 Verification Approaches

## 4.1 Verification Systems

**4.1.1 Dynamic Time Warping.** Dynamic Time Warping (DTW) is an effective method for computing the similarity between the reference and query of unequal length, by compressing or expanding to match two time series to calculate the minimal distance, which has been widely leveraged in existing verification systems [19, 31, 35, 2, 28]. Denote the reference sequence as $R : \{r_1, r_2, \cdots, r_n\}$, the query sequence as $S : \{s_1, s_2, \cdots, s_m\}$, and the warping path as $W : \{\omega_1, \omega_2, \cdots, \omega_k\}$, where k ranges from $max(n, m)$ to $m + n - 1$. Then DTW is computed as follows:

$$DTW(R, S) = min(\frac{\sqrt{\sum_{k=1}^{K} \omega_k}}{K}) \tag{1}$$

In experiments, we extract 12 time functions from online time series using the original $x, y$ coordinates, and pressure as features and input them into DTW as one of the baselines, and denote it as DTW. The details of time functions are presented in the supplementary materials.

**4.1.2 Sig2Vec.** Sig2Vec [21] is a one-dimensional CNN-based model with multi-head attention, which has achieved state-of-the-art performance on DeepSignDB [40]. The backbone network consists of the convolutional, SELU[16], and max-pooling layers. After the backbone, two selective pooling (SP) modules are employed to pool the feature sequence output by the convolutional layers into a

fixed-length vector. In this paper, it is trained under triplet loss [45] and label smoothing cross-entropy loss [34] for 200 epochs with an initial learning rate of 0.001.

**4.1.3 TA-RNNs.** The Time-Aligned Recurrent Neural Networks (TA-RNNs) [39] is a BiLSTM-based Siamese network that leverages DTW for sequence pre-alignment. The TA-RNNs is the benchmark model for the DeepSignDB [40] dataset. In our experiments, we train the TA-RNNs under binary cross-entropy loss for 200 epochs with an initial learning rate of 0.01 and stacked the distance-based verifier on it for inference.

**4.1.4 DsDTW.** DsDTW [13] is the latest state-of-the-art model for dynamic signature verification, which has won the ICDAR 2021 competition for online signature verification with obvious margins [42]. The DsDTW adopts a CRAN architecture to provide robust inputs for subsequent processing. Considering that DTW is not fully differentiable for its inputs, DsDTW introduces its smoothed formulation, soft-DTW, and incorporates the soft-DTW distances of signature pairs into the triplet loss [45] for end-to-end optimization. In this paper, the DsDTW is trained under triplet loss for 30 epochs with an initial learning rate of 0.01.

**4.1.5 DCNN.** DCNN [18] is a CNN-based model, which was designed for offline writer-independent signature verification. The authors novelly proposed Position-Dependent Siamese Network (PDSN) to model local similarity of different samples and used the M-way softmax loss function to classify writers' identities. In this paper, we feed static images into this network to extract offline modality features. The DCNN is trained under the combination of binary cross-entropy loss and label smoothing cross-entropy loss [34] for 200 epochs with an initial learning rate of 0.01.

## 4.2 Verifier

We adopt the distance-based verifier proposed by Lai et al. [21] to assess the performance as Equal Error Rate (EER%) of different baselines. In all experiments, we compute the $EER_{global}$ and $EER_{local}$ using a global threshold and a user-specific threshold, respectively.

## 5 Experiments

### 5.1 Experiment Protocol

To evaluate the potential of the proposed MSDS dataset in application and provide a thorough performance analysis, we designed the following experimental protocol:

- **Dataset splitation.** The experiments are separately conducted on the MSDS-ChS subset and the MSDS-TDS subset, in order to completely analyze the effectiveness of Chinese signatures and TDS on handwriting verification. For each subset, we divide users of each session into the same 202 individuals, using their samples as the training set. And the samples of the other 200 individuals are used as the testing set.
- **Training and testing strategies.** For training, we incorporate the training samples of two sessions to jointly train the models. For testing, we evaluate the baselines on single-session and across-session testing sets respectively. Testing with single-session data aims to evaluate systems' performances with limited samples, while testing with across-session data is to take into account the inter-session variation of users' handwriting and assess the systems in a more realistic scenario.
- **Impostor types.** Both skilled and random forgery are considered as impostor types. Skilled forgeries are selected from each user's own forged samples, and random forgeries are selected from the genuine samples of other users.
- **Template selection.** Different numbers of genuine templates used in testing may affect the final performance as Equal Error Rate (EER%). For the impostor type of skilled forgery, we consider testing with one to four templates, denoted as 4vs1, 3vs1, 2vs1, and 1vs1. For the impostor type of random forgery, we consider testing using four and one template. To guarantee the reproducibility of test results, all users' templates are the top $n$ samples among all their genuine samples. For example, templates used in the 4vs1 scenario are the first to the fourth genuine samples in all 20 ones.

### 5.2 Data Preporcessing

For online time series of Chinese signatures and TDS, we extracted 12 time functions as features, as illustrated in Section 4.1. For offline images, we first transform them into grayscale images and

applied the Gaussian filter algorithm to filter out noise. Next, we resize the images to a fixed size, which maintains the aspect ratio with white pixels padding until the preset size is reached to avoid distortion. The signature images are resized to the fixed size (64,192) at a ratio of 1:3, whereas the TDS images are resized to the fixed size (64,320) at a ratio of 1:5.

## 5.3 Verification Result and Analysis

Table 4 presents the Chinese signature verification results of different baselines on the MSDS-ChS dataset, and Table 5 presents the TDS verification results on the MSDS-TDS dataset. The findings are as follows.

Table 4: Chinese signature verification Equal Error Rates (EER%) of different baselines tested on the test set of MSDS-ChS. The results are displayed in the format of $EER_{global}/EER_{local}$, in which the former is computed under a global threshold and the latter is computed under a user-specific threshold.

| Session | Baseline | Skilled Forgery | | | | Random Forgery | |
|---|---|---|---|---|---|---|---|
| | | 4 vs 1 | 3 vs 1 | 2 vs 1 | 1 vs 1 | 4 vs 1 | 1 vs 1 |
| 1 | DTW | 2.31/0.40 | 2.75/0.44 | 2.69/0.53 | 13.67/4.16 | **0.00/0.00** | **0.21/0.00** |
| | Sig2Vec [21] | 1.33/0.45 | 1.83/0.35 | 3.29/0.73 | 6.76/0.63 | 0.17/0.17 | 1.32/0.17 |
| | TA-RNNs [39] | 3.49/2.71 | 3.83/2.91 | 5.18/3.83 | 6.99/2.03 | 0.17/0.09 | 0.36/0.03 |
| | DsDTW [13] | **0.91/0.05** | **0.98/0.10** | **1.44/0.13** | 4.20/0.05 | **0.00/0.00** | 0.39/0.00 |
| 2 | DTW | 3.34/0.55 | 3.49/0.63 | 4.58/0.70 | 12.83/3.29 | **0.00/0.00** | **0.28/0.01** |
| | Sig2Vec | 1.54/0.22 | 1.89/0.39 | 3.04/0.53 | 6.91/0.62 | 0.06/0.00 | 1.39/0.03 |
| | TA-RNNs | 3.04/2.40 | 3.23/2.60 | 4.02/3.20 | 7.63/2.84 | 0.13/0.02 | 0.30/0.08 |
| | DsDTW | **0.87/0.13** | **1.06/0.17** | **1.24/0.17** | 4.06/0.41 | 0.08/0.00 | 0.34/0.00 |
| 1 & 2 | DTW | 11.66/7.70 | 11.37/7.44 | 12.42/7.26 | 17.26/8.93 | **0.58/0.20** | **1.03/0.27** |
| | Sig2Vec | 9.03/4.97 | 8.78/4.92 | 9.87/5.16 | 15.10/7.27 | 1.93/0.74 | 5.09/1.18 |
| | TA-RNNs | 7.69/5.22 | 7.91/5.67 | 8.34/6.36 | 9.04/5.05 | 2.67/0.47 | 1.55/0.57 |
| | DsDTW | **5.91/2.90** | **5.69/2.90** | **5.96/2.77** | **9.58/3.99** | 0.84/0.11 | 1.87/0.17 |

Table 5: TDS verification Equal Error Rates (EER%) of different baselines tested on the test set of MSDS-TDS. The results are displayed in the format of $EER_{global}/EER_{local}$, in which the former is computed under a global threshold and the latter is computed under a user-specific threshold.

| Session | Baseline | Skilled Forgery | | | | Random Forgery | |
|---|---|---|---|---|---|---|---|
| | | 4 vs 1 | 3 vs 1 | 2 vs 1 | 1 vs 1 | 4 vs 1 | 1 vs 1 |
| 1 | DTW | 2.97/0.53 | 2.96/0.60 | 3.42/0.93 | 7.82/1.60 | **0.00/0.00** | **0.08/0.00** |
| | Sig2Vec [21] | 0.96/0.24 | 1.18/0.20 | 1.37/0.17 | 2.95/0.52 | 0.07/0.00 | 0.40/0.01 |
| | TA-RNNs [39] | 2.63/1.94 | 2.72/2.01 | 3.09/2.37 | 5.05/1.68 | 0.08/0.03 | 0.22/0.05 |
| | DsDTW [13] | **0.72/0.00** | **0.90/0.00** | **0.73/0.06** | **2.50/0.25** | **0.00/0.00** | 0.14/0.02 |
| 2 | DTW | 2.14/0.57 | 2.28/0.63 | 3.06/0.72 | 6.99/0.63 | **0.00/0.00** | **0.14/0.00** |
| | Sig2Vec | 0.51/0.09 | 0.60/0.04 | 1.06/0.09 | 1.76/0.30 | 0.06/0.01 | 0.44/0.09 |
| | TA-RNNs | 1.17/0.87 | 1.27/1.02 | 1.40/1.33 | 3.87/0.87 | 0.16/0.07 | 0.33/0.07 |
| | DsDTW | **0.42/0.11** | **0.35/0.09** | **0.36/0.11** | **2.02/0.28** | 0.01/0.00 | 0.37/0.03 |
| 1 & 2 | DTW | 9.99/5.75 | 9.94/5.78 | 10.01/5.95 | 14.46/6.76 | **0.25/0.01** | **0.30/0.04** |
| | Sig2Vec | 5.18/2.07 | 5.24/2.22 | 5.94/2.17 | 7.01/3.26 | 1.66/0.26 | 1.76/0.28 |
| | TA-RNNs | 5.11/2.91 | 5.44/3.06 | 5.77/3.16 | 5.94/2.60 | 1.71/0.40 | 0.85/0.21 |
| | DsDTW | **4.13/1.42** | **4.05/1.41** | **4.40/1.32** | **5.76/1.85** | 0.42/0.07 | 0.59/0.14 |

**Across-session performance degradation.** From Tables 4 and 5, it is obvious that all models achieve better performance when tested with single-session data than with across-session data. When performing handwriting in a single session, users can maintain their writing styles within smaller variances, bringing less verification difficulty. After the time gap of at least 21 days, their handwriting styles can drastically change, reasonably increasing the intra-class variance. Hence, when the templates and queries are from different sessions, models' performances degrade. This puts in evidence that the inter-session variation is a key factor that should be considered in real-world applications for verification systems.

**Under different scenarios.** Among the studied systems, DsDTW [13] outperforms others under attacks from skilled forgeries, and DTW yields the best performance under attacks from random forgeries. In single-session scenarios, Sig2Vec [21] can also achieve satisfactory performances. Therefore, robust systems that perform well both in skilled and random forgery scenarios remain to be explored, and this paper has contributed a large-scale MSDS dataset for promoting such kind of research.

**Using different numbers of templates.** According to Tables 4 and 5, the optimal verification result for Chinese signatures and TDS may occur when the number of templates is set as 3 or 4. Owing to the existence of intra-class variance, the verification performance and number of templates are not positively correlated. This observation is consistent with the experimental results for digit combinations with different lengths in [37].

**Effectiveness of different data.** Tables 4 and 5 suggest that all models perform better on MSDS-TDS than on MSDS-ChS. Note that the two subsets are simultaneously collected from the same users. This finding is inspiring that the accuracy of TDS verification is higher than that of Chinese signature verification under the same conditions. The reasons may lie in two aspects. First, handwriting styles in TDS may be easier to be discovered and learned owing to the more sparse spatial architectures than Chinese signatures. Second, compared with Chinese signatures, Token Digit Strings are more difficult to be imitated for they include more unique patterns generated by personal writing habits, such as spacing and skew, which indicates that the genuine and forged TDS are easier to be distinguished. Therefore, it is highly recommended to adopt Token Digit Strings, like phone numbers, ID card numbers, and other distinctive personal numbers as handwritten codes for high-accurate online identity verification as a better option for Chinese signatures.

**Failure verification analysis.** We collect several incorrectly verified samples as shown in Figure 4, including the false accepted and false rejected ones. For false acceptances, they are mostly skilled forgeries of high forgery quality and can not be easily distinguished. For false rejections, although they are genuine samples of the claimed user, they differ significantly from the templates owing to the inter-session variation, resulting in false classification into forged samples.

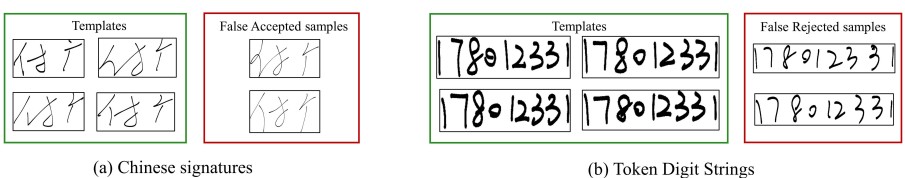

(a) Chinese signatures          (b) Token Digit Strings

Figure 4: False accepted samples and False rejected samples.

## 5.4 Modality Fusion

Table 6 demonstrates the results of fusing the data of online and offline modalities. We feed online time series into Sig2Vec [21] and offline images into DCNN [18], then concatenate the output vectors of the two models as new feature representations. The new features are then input to the verifier mentioned in Section 4.2 to make verifications. From the third and fifth rows, it can be observed that on the MSDS-ChS and MSDS-TDS datasets, the EER% is improved when the offline information is added. This is because static images possess features that dynamic time series lacks, such as the global relationship in spatial architectures of handwriting and local relationship between different strokes. Hence, the feature representation would be enhanced to be more distinguishable by combining the online and offline features, resulting in better performances of handwriting verification systems.

Table 6: Experimental results of fusing the online and offline modality data.

| Subset | Modality | | Skilled Forgery | | Random Forgery | |
|---|---|---|---|---|---|---|
| | online | offline | 4vs1 | 1vs1 | 4vs1 | 1vs1 |
| MSDS-ChS | ✓ | ✕ | 9.03/4.97 | 15.10/7.27 | 1.93/0.74 | 5.09/**1.18** |
| | ✓ | ✓ | **8.44/4.07** | **14.98/6.38** | **1.71/0.60** | **3.54**/1.53 |
| MSDS-TDS | ✓ | ✕ | 5.18/2.07 | 7.01/3.26 | 1.66/0.26 | 1.76/**0.28** |
| | ✓ | ✓ | **4.95/1.88** | **6.98/3.22** | **1.43/0.17** | **1.62**/0.53 |

## 5.5 Cross Dataset Validation

We also conduct a cross-dataset evaluation on DeepSignDB [40] and MSDS-ChS using the optimal DsDTW [13] model. The cross-dataset evaluation shows that DsDTW trained on DeepSignDB behaves poorly on the Chinese signature dataset in skilled forgery scenarios due to the domain shift, so it is necessary to collect a large-scale dataset for Chinese signature verification. Surprisingly, when tested on MSDS-ChS in random forgery scenarios, the DsDTW trained on western DeepSignDB (with 528 training users) outperforms that trained on MSDS-ChS (with 202 training users), indicating that increasing the user capacity of the training set is beneficial for the deep model to better distinguish random forgeries. In addition, when trained on the MSDS-ChS dataset, DsDTW delivers better verification performance on DeepSignDB than on MSDS-ChS, proving that the proposed MSDS-ChS dataset is more challenging. This is because Chinese signatures are usually composed of multiple discrete strokes with larger intra-class variance, unlike writing-friendly cursives commonly seen in western signatures.

Table 7: Cross-dataset validation results on MSDS-ChS and DeepSignDB [40] using the optimal DsDTW model.

| Training Set | Testing Set | Skilled Forgery | | Random Forgery | |
|---|---|---|---|---|---|
| | | 4vs1 | 1vs1 | 4vs1 | 1vs1 |
| DeepSignDB [40] | DeepSignDB | **2.54/0.92** | **4.04/1.50** | 0.97/0.19 | 1.69/0.57 |
| MSDS-ChS | DeepSignDB | 4.77/2.24 | 9.09/3.30 | 2.76/0.98 | 4.62/1.86 |
| DeepSignDB | MSDS-ChS | 10.60/5.78 | 14.63/6.08 | **0.41/0.06** | **0.53/0.05** |
| MSDS-ChS | MSDS-ChS | 5.91/2.90 | 9.58/3.99 | 0.84/0.11 | 1.87/0.17 |

# 6 Limitations

The offline handwriting images in the MSDS dataset are rendered on the acquisition devices while collecting online information, rather than being acquired by typically photographing or scanning (e.g. [15, 46, 44]). Therefore, the rendered handwriting may differ from the one written by pens in terms of tips, turning points, and thickness of the strokes, which may lead to changes in personal handwriting information.

# 7 Conclusion

In this paper, we propose the new MSDS dataset, consisting of two subsets: MSDS-ChS and MSDS-TDS. They were simultaneously collected according to the same manner, contributed by 402 users with 20 genuine samples and 20 skilled forgeries per user per subset. The data were acquired in two sessions with a time interval of at least 21 days. To the best of our knowledge, MSDS-ChS is the largest publicly available handwritten Chinese signature dataset. MSDS-TDS novelly covers Token Digit Strings (TDS), i.e. the actual phone numbers of users, which have not yet been investigated, and can serve as a new benchmark dataset to facilitate related research. The two subsets are provided in both online modality with time series and offline modality with rendered images.

We conduct a thorough benchmark evaluation of MSDS on multiple baselines and perform a comprehensive analysis. Verification performances on MSDS-TDS generally outperform those on MSDS-ChS, which reveals that the handwritten Token Digit String could be a potentially more powerful biometric than handwritten Chinese signatures.

For future work, we expect to design more specified models to reduce the EER% on both subsets in across-session scenarios. It would also be worthwhile to exploit handwritten Chinese signatures and Token Digit Strings together in an across-modality manner to explore the model's performance and its feasibility for real-world applications. Additionally, regarding data diversity, we will consider collecting more handwriting from different age groups and various countries to further enrich the age and regional diversity of our dataset if conditions permit.

## License

The MSDS dataset should be used under Creative Attribution-NonCommercial-NoDerivatives 4.0 International (CC BY-NC-ND 4.0) License for non-commercial research purposes.

## Acknowledgement

This research is supported in part by NSFC (Grant No.: 61936003), GD-NSF (no.2017A030312006, No.2021A1515011870), Zhuhai Industry Core and Key Technology Research Project (no. ZH22044702200058PJL), and the Science and Technology Foundation of Guangzhou Huangpu Development District (Grant 2020GH17).

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
