# OpenReview forum: "MSDS: A Large-Scale Chinese Signature and Token Digit String Dataset for Handwriting Verification"
_NeurIPS.cc/2022/Track/Datasets_and_Benchmarks — NeurIPS 2022 Datasets and Benchmarks _

### Official Review · Reviewer_AbXT · 2022-07-08
**A fair comparison between chinese signatures and token digit strings corpus.**

**Rating:** 8
**Confidence:** 5

**Strengths:**

This publicly available dataset is, by its size and level of granularity, the biggest one available for chinese signature at the time I am writing this. It's also including multiples modalities, which are very useful for modern online verification tools like the one's presented in the up-to-date "approaches" section. The way how the dataset as been collected make it even more unique of its kind, the number of users involved in the campaign, hardware variety (technically and financially), the fact that you make the users switch their writing hand for three forged samples and the delay between each session is way beyond the average datasets currently available on the market. The data distribution and collection across both tasks is perfect and open new field of research for identity verification based on handwriting outside the chinese domain. Authors also performs appropriate state-of-the-art experimentations that define a good baseline for future works on this dataset. I found it very appropriate to see the modality fusion and was surprised of the outcomes, we generally can expect higher performances from online architecture but never see other papers combining both at the same time for the task.

**Weaknesses:**

We don't have any information about the users genders, ages, level of education, permanent disease, main hand or even if they wear glasses ? Such information can be extremely useful for others tasks and doesn't cost very much to get. It can also put in evidence some bias during the data collection.

About hardware, it could be nice to have the identifier of the device used to collect a specific signature or TDS, it will allow studying inter and intra-device variability. Also, maybe some users are already familiar with one of the two tablets before the event and have an ergonomic advantage compare to others. We don't have any information about if the users have to switch between devices during data collection, or if everybody went through the same devices in the same order ? If users have been changed of tablets models between the first and second session, this can, among other things, explain the inter-session variability. The screen real estate isn't exactly the same, and stylus ergonomic differ from one to another.

One missing reference at line 173, "obvious margins[?].".

You can also add a reference for the DTW, according to the Wikipedia page of the DTW, one of the first reference is : Vintsyuk, T. K. (1968). "Speech discrimination by dynamic programming". Kibernetika. 4: 81–88.

**Additional Feedback:**

The tremendous amount of work put in place to collect such multi-modal dataset and analyzing inter-session variability must be rewarded. This work open new field of possibilities for identity verification using the TDS rather than signature as we can generally see in the literature. Combining modalities using fusing is also a new perspective to improve performances and accuracy of these systems.

I would like to see this dataset released fully open-source and see how the community could grab it through various challenges like we can see on Kaggle. However, this could be dangerous in terms of privacy, since signatures contains first and last names of the users.

I also would like to know why you have not incorporate in the dataset more information about the context like the device used, user gender and age ? Have you done it to preserve user's personal information ? Or, it's just something you have not thought about ?

**Clarity:**

Well organized, written with a good english, the paper can be read by anyone without any requirements and the tables are extremely clear.

The only thing I would like to have is a description of the Equal Error Rate (EER%) and how it's computed, since the paper have to be open to a broad range of peoples, sometimes not familiar with the biometrics and such addition wouldn't be too much.

**Correctness:**

The metric chosen to evaluate the task is one generally used for similar datasets and the experiments are based on a vast variety of architectures, from the historical DTW to the state-of-the-art DsDTW and represent correctly the approaches that could be used to perform the task.

**Documentation:**

For the dataset collection and organization, it's very clear. For the availability, the authors put their effort in releasing the dataset on GitHub with a nice description. But, when I take a look at the GitHub repository, I noticed that the licensing is not available in the paper, it will be simpler for the papers to follow and users interested in the project to get this information in the paper and not digging to get the information.

It will be very cool to get access to the corpus on other platform like Zenodo to attract even more peoples to the task and allow a better growth from the performances point of view. The maintenance plan is clear on the GitHub page.

From the ethical and responsible point of view, nothing to complain about this. The user's anonymity cannot be preserved in such task, but the purpose of this task seems to be devoid of bad intentions. The only risk is to have badly intentioned peoples to have access to the dataset without any permission and use these signatures to impersonate users identities. So to prevent it to happen, it would be good to change the password of the archive to one even stronger composed of specials characters and longer.

The approaches used during the benchmarks section have clear parameters and allow a good reproducibility.


**Ethics:**

Does the dataset use features or label information about individual names?
> It contains genuine signature of the users, so we can link this signature to the authors as is.

Did people provide their consent on the collection of such data?
> Yes, they have a specific number of tasks to perform during the collection.

Could the use of the data be degrading or embarrassing for some people?
> No, no personnal information that can be used to dregradating peoples image or affect mental health of the peoples.

Contain information that could be deduced about individuals that they have not consented to share ?
> No.

Encode, contain, or potentially exacerbate bias against people of a certain gender, race, sexuality, or who have other protected characteristics ?
> No.

Contain human subject experimentation and whether it has been reviewed and approved by a relevant oversight board ?
> No.

Have been discredited by the creators ?
> No.

**Relation To Prior Work:**

Absolutely clear, the authors take their time to describe the datasets available for the task during the section 2, tables 2 and 3.

However, some recent works are missing in the table 2 : During CVPR 2022, a new dataset called "ChiSig" and presented in the paper called "Signature Detection, Restoration, and Verification: A Novel Chinese Document Signature Forgery Detection Benchmark" was released and offers, among other things, a chinese signature verification task. This paper also refers, during the table 1, to a dataset from 2017 called CSD presented in the paper "Offline Signature Verification Using Local Features and Decision Trees" by Juan Hu, Zhenhua Guo, Zhenyin Fan and Youbin Chen and which containing 5,400 signatures from 300 users. They also refer to another one, containing, this time, 29,000 signatures from 749 users in the paper "Inverse Discriminative Networks for Handwritten Signature Verification" presented this time during CVPR 2019 by Ping Wei, Huan Li and Ping Hu. I can under why you missed them, the recent one is publicly available but is released recently, so I can understand why he is not part of your work. For the others, their access is limited but still present in the literature and require some kind of references in your paper.

**Summary And Contributions:**

During this paper, the authors has presented a new signature and token digit string dataset collected from the same 402 chinese participants during two sessions spaced by an interval of 21 days. This dataset has the advantage of combining both online and offline modalities in such a large quantity, that make it one's of this kind. This dataset was collected in a controlled context in such a way that inter-session variability can be measured. Authors also evaluate this dataset performance using classical and state-of-the-art verification approaches in various scenarios for highlighting the limits of them in more real-world cases. They also prove the effectiveness of using other type of biometric such as handwritten token digit strings from users phone numbers or ID card numbers for a more accurate identity verification. To conclude, they apply cross-dataset validation with the western signature dataset DeepSignDB to understand if such dataset can be more effective in specific scenario like random forgery attacks.

---

### Official Review · Reviewer_ETqJ · 2022-07-22
**Large and useful dataset for forgery detection within signatures and token digit strings**

**Rating:** 7
**Confidence:** 3
**Correctness:** The evaluation methods and experiment…
**Clarity:** The paper is mostly clear and well wr…

**Strengths:**

1. This paper presents a new dataset of Chinese signatures and token digit strings, and it is the largest of its type. This dataset is going to be useful for research in forgery prevention via machine learning models.
2. The authors comprehensively compared their new dataset with existing ones, and clearly listed the novelties or advantages of their dataset in Table 2 and 3.
3. The authors benchmarked their dataset with sufficient number of recent machine learning models on signature forgery detection, and have the surprising finding that the models were able to better detect forgery from token digit strings than Chinese signatures. Moreover, they conducted additional experiments to show that including both online and offline modalities help, and they also cross-validated their dataset with DeepSignDB.

**Weaknesses:**

1. There lacks documentation on a few key parts of the data collection process. How are the participants selected? How were consent obtained from the participants since the data collected involves their names and phone numbers and is released publicly?

**Additional Feedback:**

N/A

**Documentation:**

The procedure of collecting signatures and token strings are well documented.
However, there lacks documentation on how the participants are selected, how the authors obtained the participants' consent to publicly release their signature and phone number, whether the participants are compensated, etc.

**Ethics:**

The dataset includes personally identifiable information, including the names of the 402 users and their current or previous phone numbers. It is unclear whether and how the authors have obtained consent from participants.

**Relation To Prior Work:**

The authors very clearly discussed how this work differs from previous contributions. They comprehensively compared their datasets to existing datasets, and summarized the comparisons in Table 2 and 3.

**Summary And Contributions:**

This paper presents a new dataset, MSDS, that contains two parts: Chinese signatures and their forgeries, and token digit strings and forgeries. The dataset is the largest of its type by far, and also provides both the online and offline modalities of each signature and token digit string. The authors comprehensively benchmarked the datasets with existing SOTA models.

---

### Official Review · Reviewer_UaB1 · 2022-07-24
**Review of "MSDS: A Large-Scale Chinese Signature and Token Digit String Dataset for Handwriting Verification"**

**Rating:** 5
**Confidence:** 2
**Correctness:** Yes, the evaluation and the data coll…
**Clarity:** Yes, it is clearly written.

**Strengths:**

1. The paper is well-organized, and the proposed dataset is large in scale, containing both genuine and skilled forgeries samples.
2. The paper sheds light on the importance of the Token Digit String instead of separate digits and can assist future verification research on the writers' inter-session variation.
3. The authors have performed thorough experiments on the proposed dataset and bring a new perspective on the potential use of the Token Digit String as a better biometric for handwritten verification.
4. The dataset has multiple modalities, including static images, coordinates, pressure, and time stamps.

**Weaknesses:**

I am not an expert on handwritten signature verification, but I have some major concerns about this work:
1. I don't see whether this topic is suitable for NeurIPS Track Datasets and Benchmarks; maybe venues like biometric authentication (where previous works were published) are better for this work?
2. Where is the checklist?
3. Ethical concerns: As the authors claim, this dataset contains the users' real names and phone numbers. I don't see any explicit discussion about the ethical considerations of the collection process: for instance, did people provide their consent on the collection of such data (General Ethical Conduct)?

**Additional Feedback:**

The title has a typo: "Handwrting" -> "Handwriting".
Table 2: What does Z denote?


**Documentation:**

The documentation, GitHub repo, and URL are good. The maintenance plan is discussed in the supplementary. My only concern is the lack of the requested checklist.

**Ethics:**

As the authors claim, this dataset contains the users' real names and phone numbers. I don't see any explicit discussion about the ethical concerns of the collection process: for instance, did people provide their consent on the collection of such data (General Ethical Conduct)?

**Relation To Prior Work:**

I think the authors discuss thoroughly how this work differs from previous work and employ several tables for comparison. But I am not an expert in this field, so it is unclear to me.

**Summary And Contributions:**

This paper presents a new Chinese handwriting verification dataset containing handwritten Chinese characters and digits. It is the largest publicly available Chinese signature dataset. The authors performed experiments on both MSDS-ChS and MSDS-TDS, showing that the handwritten token digit string is a better biometric indicator for handwriting.

---

### Official Review · Reviewer_UrgJ · 2022-07-24
**Review of "MSDS: A Large-Scale Chinese Signature and Token Digit String Dataset for Handwriting Verification"**

**Rating:** 7
**Confidence:** 2
**Clarity:** The paper is well-written and clear o…

**Strengths:**

* MSDS-ChS is the largest publicly available handwritten Chinese signature dataset.
* MSDS-TDS novelly covers the actual phone numbers of users, which has not yet been investigated.
* Thorough and extensive benchmark to provide baseline results
* Both subsets are provided in both online and offline modalities.

**Weaknesses:**

* Accessibility of the data: the MSDS dataset can only be used for non-commercial research purposes. The authors require the potential users to submit an application form and provide a recent publication list in order to obtain the decompression password, which may potentially introduce entry barriers for young researchers, researchers with conflict of interest, etc. What is the motivation of requiring this verification? If this verification is indeed necessary for any reasons, the authors should elaborate more on the process about how they plan to approve the application forms in a transparent and objective way. The authors should also promise how much time this verification service will be maintained.
* The dataset contains signatures, thus names of the data contributors, which is personally identifiable information and can reveal identities. The authors should discuss the privacy issues. How will the data contributors' identities be protected? Did data contributors provide their consent to use or share the data? Did they explicitly know and the purpose of the collection of such data and potential risks? Will the misuse of such data lead to any potential negative societal impact? If yes, do the authors have any measures to fight against that? Will individual’s right to be forgotten (removed from the dataset; GDPR) be ensured?
* The [NeurIPS checklist](https://neurips.cc/public/guides/PaperChecklist) is missing.



**Additional Feedback:**

* The paragraph from Line 243 to Line 253. Are there any tentative explanation of this surprising experimental results (MSDS-TDS are generally better than those on MSDS-ChS)? Does this result correspond to results in related works? Do western signatures (which consists of common characters, like digits) provide better results than Chinese signatures?
* How long will the link to download data remain available? How long will be dataset be maintained?
* More details of how the maintenance is supposed to work will be appreciated. What is the mechanism of data correction/versioning?


Typo:

Title: Handwrting => Handwriting
 Line 66: seperate = > separate
Line 173: a missing reference


**Correctness:**

The dataset is constructed in a sound way (as far as I can tell). The evaluation methods and experiment design look good.

**Documentation:**

The documentation is good enough. Sufficient details are given.

**Ethics:**

Signatures can be considered human-derived data and reveal the names of data contributors, they are also a common biometric measure exploited in person verification. There might be issues related to the privacy of the data contributors. Please refer to Weaknesses for the details.

**Relation To Prior Work:**

This paper clearly discussed how this work differs from previous contributions.

**Summary And Contributions:**

This paper introduced a new handwriting verification benchmark dataset named MSDS. It consists of two subsets: MSDS-ChS for Chinese Signatures, MSDS-TDS for Token Digit Strings. This dataset was contributed by 402 data contributors with 20 genuine samples and 20 skilled forgeries per user in each subset. The authors verified the usefulness of the proposed dataset with extensive experiments.

---

### Official Review · Reviewer_PPfu · 2022-07-24
**In general, it is a good dataset paper. But the authors are suggested to conduct more polishing.**

**Rating:** 6
**Confidence:** 3
**Correctness:** The evaluation methods and experiment…

**Strengths:**

1. This paper proposed a new dataset for an important task -- handwriting verification.
2. This paper provided the sufficient details e.g., the process of collecting this dataset, the SOTA baseline experimental results.


**Weaknesses:**

1. The size of collected dataset is small (402 * 20) for the time series and image tasks. The current SOTA neural networks need the large size of datasets to train and can then make some convincing conclusions. If the cost is not high, the authors are suggested to collect more data. But I can change my opinion if stronger arguments are provided by the authors.

2. This paper needs more polishing. The details are provided in the “Clarity” part.


**Additional Feedback:**

I have one question. Why splitting the training/test dataset 50%/50% instead of 80%/20% or 90%/10% which seem more common in the machine learning tasks?

**Clarity:**

This paper is generally well-written. But it needs further polishing. Several examples are as follows:
1. In line 113, the authors said, “…in their future work…”. If the paper was published, the authors are suggested to say “…In their following/next/another work…” instead of “…in their future work…”.
2. There are inconsistent citation formats. In line 81 “…In 2003, Ortega et al. [26] published the MCYT database…”, line 83 “Yeung et al. proposed the SVC [39] dataset”, line 98 “…Kalera et al. [13] proposed the CEDAR dataset…”, and line 115 “…Chen et al. [3] proposed the LERID database…”, the authors are suggested to make these citation formats consistent. For example, you can change line 83 “Yeung et al. proposed the SVC [39] dataset” to “Yeung et al. [39] proposed the SVC dataset”.
3. Although the authors explain the difference between online and offline samples in the supplementary material, the pointer is suggested to be added in the main paper. I keep being confused until I read section 5.4 in the main paper and supplementary material.
4. It is wordy for some sentences. For example, in line 141 “…20 genuine signatures, 20 genuine phone numbers, 20 forged signatures, and 20 forged phone numbers…”, it is a little wordy.
5. The authors are suggested to provide a simple description in the caption or footnote on what EER_{global} and EER_{local} mean in Table 4 & 5 even though you explained them in the body.
6. There is a citation error in line 173 “…margins[?]…”.


**Documentation:**

This paper provided the sufficient details on documentation.

**Ethics:**

An obvious concern is that some people can misuse this kind of signature dataset proposed by the authors to learn generating the skilled forgeries of the people in the real world. The authors are suggested to add a discussion about the ethics issues.

**Relation To Prior Work:**

This paper clearly described the relations to prior work.

**Summary And Contributions:**

This paper proposed a new Multi-modal Signature and Digit String (MSDS) dataset including two subsets: MSDS-ChS (Chinese Signatures) and MSDS-TDS (Token Digit Strings), which is the largest publicly available Chinese signature dataset. The authors also provided the baseline experimental results using the SOTA methods and found the interesting result that the models utilizing Token Digit String perform better than the ones on Handwritten Chinese signature.

---

### Review · Ethics_Reviewer_5zYx · 2022-08-22

**Recommendation:** 2

**Ethics Documentation:**

In my opinion, the authors provide all sufficient information on data organization, availability, and maintenance.


The authors do not provide sufficient information on the ethical and legal aspects of the data collection: The dataset contains personal information about individuals, and the authors claim that the individuals provided their consent to share their personal information and that individuals have the right to be forgotten. The authors do not provide the corresponding forms. Also, the mechanism for fulfilling the right to be forgotten is not clear. (This issue was also partially raised by Reviewer UrgJ).


The authors provide their dataset for non-commercial research purposes and under the condition of filling out an application form (https://github.com/sincert/MSDS/blob/main/application-form/Application-Form-for-Using-MSDS.docx).


The authors distribute their dataset under the CC BY-NC-ND 4.0 license.

**Ethics Review:**

The authors propose a novel dataset of Chinese Signatures (ChS) and Token Digital Signatures (TDS). The signatures and token digital signatures (phone numbers) were collected from 440 individuals.


The paper contains some general ethical issues (https://neurips.cc/public/EthicsGuidelines), which I describe below:
1. The dataset contains personally identifiable information about individuals (the genuine signatures and the actual phone numbers).
2. The TDS and the ChS data were collected from the same set of individuals (who presumably can write in Chinese), which raises the question of the regional diversity of the TDS dataset. To my knowledge, the handwriting of digits can hugely vary with the regions (https://en.wikipedia.org/wiki/Regional_handwriting_variation); hence, the applicability of the resulting TDS dataset seems limited.


Additionally, since there is no information provided on the diversity of individuals about which the data was collected, it is not clear how diverse would be the set of Chinese signatures (ChS). I assume there should also be age/regional diversity in written Chinese.


I could not find any discussion on the potential negative societal impact of the paper. For example, as mentioned by Reviewer PPfu, the dataset can be used for generating realistic forgeries. The authors are encouraged to mention this issue in the paper and to discuss how it can be mitigated.

---

### Meta-Review · Area_Chair_MARh · 2022-09-09

**Recommendation:** Accept
**Confidence:** 4

**Metareview:**

The paper gets divergent reviews initially. The reviewers appeciate the largest publicly available scale of the dataset for handwriting verification, novel data collection process, extensive experimental validation on baselines and modality fusion, and findings on the importance of Token Digit String. The main concerns include ethics due to personally identifiable information, incomplete information on users and devices, unavailable license, missing checklist, and unclear presentation.

The rebuttal is successful at addressing many of these concerns. Majority of the reviewers are satisfied and support acceptance post-rebuttal. The main remaining concern is on the ethics. The authors have taken extra care and measures to ensure the personally identifiable information is not wrongly used. However, the limited age/regional diversity remains a serious issue.

The ACs agree with the majority assessment on the contributions of this paper, and take into account of the unique challenge of collecting handwriting verification data at scale, and thus recommend acceptance.

In the final version, the authors should add explicit discussion on the dataset's limitations on age/regional diversity, and improve clarity on other issues raised by the reviewers.

---

### Decision · Program_Chairs · 2022-09-16

Accept